# Evaluation of Clinical and Laboratory Characteristics of Children with Pulmonary and Extrapulmonary Tuberculosis

**DOI:** 10.3390/medicina55080428

**Published:** 2019-08-01

**Authors:** Deniz Aygun, Necla Akcakaya, Haluk Cokugras, Yıldız Camcıoglu

**Affiliations:** Department of Pediatric Infectious Disease, Cerrahpasa Medical School, Istanbul University-Cerrahpasa, 34303 Istanbul, Turkey

**Keywords:** children, extrapulmonary, pulmonary, tuberculosis

## Abstract

*Background and objective:* Tuberculosis (TB) is an important public health problem in both developing and developed countries. Childhood TB is also an important epidemiological indicator in terms of forming the future TB pool. The diagnosis of TB is difficult in children due to the lack of a standard clinical and radiological description. We aimed to evaluate and compare the clinical, laboratory, and radiologic findings of childhood pulmonary and extrapulmonary TB. *Material and Methods:* The medical records of patients hospitalized with the diagnosis of pulmonary tuberculosis (PTB) and extrapulmonary tuberculosis (EPTB) between December 2007 and December 2017 were evaluated retrospectively. *Results*: There were 163 patients diagnosed with TB with 94 females (57.7%) and 69 males (42.3%). Seventy-three patients (44.8%) had PTB, 71 (43.6%) patients had EPTB, and 19 patients (11.7%) had both PTB and EPTB, called as disseminated TB. Ninety-six (58.9%) patients had tuberculin skin test (TST) positivity and 64 patients (39.3%) had interferon-gamma release assay (IGRA) positivity. Acid-resistant bacteria were observed in 34 (20.9%) body fluid samples and culture positivity was observed in 33 (20.2%) samples. Comparison of PTB, EPTB, and disseminated TB revealed that low socioeconomic status, TB contact, and low body weight were more common in disseminated TB, and TST positivity was more common in PTB. *Conclusion*: Malnutrition, low socioeconomic status, and TB contact were important diagnostic variables in our study and all three parameters were more common in disseminated TB. Tuberculosis should be considered in patients admitted with different complaints and signs in populations with high TB incidence and low socioeconomic status.

## 1. Introduction

Tuberculosis (TB) is an important public health problem in both developing and developed countries due to immigration; despite control strategies. In 2017, the World Health Organization (WHO) reported an estimated 10.0 million TB cases and 1 million of these were children [1]. Tuberculosis is a highly contagious and mortal disease. Although the vaccine against *Mycobacterium tuberculosis* was developed many years ago, TB remains the most common cause of death from a single infectious agent [2]. In 2017, 1.3 million people died due to TB [1]. Turkey has an intermediate prevalence of TB, with an incidence rate of 17 per 100,000 inhabitants [1].

Although the TB bacilli are prone to pulmonary involvement, they may be retained in almost all tissues and organs without differentiation. As the lymphohematogenous spread is higher in children, the progression of infection into disease and extrapulmonary involvement is more common [3]. Moreover, young children, especially younger than 2 years old, are more prone to develop active TB [4]. Childhood TB is a health index that reflects uncontrolled adult TB in the population. Childhood TB is also an important epidemiological indicator in terms of forming the future TB pool. The accurate diagnosis of the disease is important not only for effective treatment but also for the detection of contact cases of the infected child. However, diagnosis of TB is difficult in children due to the lack of a standard clinical and radiological description [5,6]. The microbiological confirmation rate is also low in children due to technical difficulties in sampling [7]. Diagnosis is usually made by a combination of history and tuberculin skin test (TST) and radiology findings. The data established for adults is generally used in diagnosis and treatment of tuberculosis in pediatric patients.

We, therefore, evaluated and compared the clinical, laboratory, and radiologic findings of childhood pulmonary and extrapulmonary TB.

## 2. Material and Method

### 2.1. Study Design

This retrospective single-center study enrolled patients at Istanbul University Cerrahpasa Medical Faculty Department of Pediatric Infectious Disease between December 2007 and December 2017. We have tuberculosis outpatient clinic and well-maintained data of our patients. The detailed data of 163 patients diagnosed with TB were analyzed. Istanbul is the most populous city in Turkey and also has the highest incidence of TB.

The medical records of patients were reviewed to obtain data on patient age, sex, bacille Calmette–Guérin (BCG) vaccination status, underlying diseases, contact history, TST results, or interferon-gamma release assay (IGRA) positivity, laboratory and radiologic findings, *M. tuberculosis* culture, and treatment of choice and complications.

### 2.2. Patient Definitions

The diagnosis of TB was made through a history of exposure to a positive case; positive TST or IGRA, clinical, radiological, and histopathological findings; and microbiological results. Patients were defined as confirmed TB cases when they presented a positive microbiological evaluation either with direct microscopic evaluation or *M. tuberculosis* culture from an adequate sample.

In the absence of microbiologic confirmation, the patients were considered to be probable TB if they had (1) signs or symptoms of TB, (2) immunological evidence of *M. tuberculosis* with positive TST or IGRA findings, and (3) treatment with two or more anti-TB drugs and a complete diagnostic evaluation [8,9].

Pulmonary TB (PTB) was defined as any bacteriologically confirmed or clinically diagnosed case of TB involving the lung parenchyma or the tracheobronchial tree. Extrapulmonary tuberculosis (EPTB) was defined as any confirmed or probable TB case involving organs other than the lungs, pleura, or intrathoracic lymph nodes [10]. Patients with both PTB and EPTB were classified as disseminated TB.

All three groups were compared. The collected variables included age, sex, BCG vaccination status, body weight, socioeconomic status, contact history, comorbid disorders, TST and IGRA positivity, radiological and histopathological findings, culture positivity, and drug resistance.

### 2.3. Laboratory Investigations

A standard TST was performed by placing purified protein derivative (PPD) (0.1 mL containing 5 Todd units) intradermally and reading the results after 48–72 h. In children with a BCG scar, an induration of ≥15 mm was considered a positive reaction. An induration of ≥10 mm was classified as a positive result in unvaccinated children [11].

We have special microbiology department only for TB in our clinic and a specialist studies the samples. Clinical samples including sputum, gastric aspirate, cerebrospinal fluid, pleural fluid, urine, and biopsy materials were examined by Ziehl–Neelsen acid-fast staining. Each sample was inoculated onto Löwenstein–Jensen slants as the stand-alone medium and then incubated at 37 °C for eight weeks. The histopathologic analysis of the tissue specimens was also recorded.

The posterior-anterior (PA) chest radiograph and/or thoracic computerized tomography findings of all patients with PTB were evaluated for hilar or mediastinal lymphadenopathy, consolidation, atelectasis, pleural effusions, and miliary TB. Computerized tomography was considered for gastrointestinal tract, genitourinary, and renal TB, and either computerized tomography or magnetic resonance imaging was considered for central nervous system TB along with X-rays.

### 2.4. Statistical Analysis

IBM SPSS Statistics for Windows, version 21.0 (IBM Corp., Armonk, NY, USA), was used for statistical analysis. Numerical data were expressed as mean ± standard deviation and categorical data were expressed as frequency (n) and percentage (%). Shapiro–Wilk tests were used to show continuous measurement load. X^2^ tests were used to compare the differences. Mann–Whitney tests were used for the analysis of nonparametric values.

Approval was obtained from the Cerrahpasa Medical Faculty Local Ethics Committee (14.03.2018-29430533). A signed informed consent form was obtained from all parents.

## 3. Results

### 3.1. Demographic Data

A total of 163 patients hospitalized at the Pediatric Infectious Disease Department were enrolled in the present study. Ninety-four (57.7%) of patients were female and 69 (42.3%) were male. Their mean age was 9.85 ± 5.03 years (0.25–17 years). Seventy-three patients (44.8%) had PTB, 71 (43.6%) patients had EPTB, and 19 patients (11.7%) had both PTB and EPTB. The EPTB cases included 29 (32.2%) patients with TB lymphadenitis, 14 (15.6%) with gastrointestinal TB, 12 (13.3%) with TB meningitis, 12 (13.3%) with miliary TB, 10 (11.1%) with pleural TB, six (6.7%) with bone TB, three (3.3%) with renal TB, two (2.2%) with skin TB, one (1.1%) with genitourinary TB, and one (1.1%) with pericardial TB. Totally renal, skin, genitourinary, and pericardial TB were classified as others (7.7%). The demographic data of patients are shown in Table 1.

Ninety-seven (59.5%) families had a low socioeconomic status. The body weights of 80 patients (49.1%) were below the third percentile. Twenty (12.3%) patients had comorbid disease. Ten of the patients had congenital immunodeficiency (five with chronic granulomatous disease, three with hypogammaglobinemia, and two with interferon-gamma receptor deficiency). Three of the remaining seven patients had juvenile idiopathic arthritis, one had inflammatory bowel disease, and one had chronic renal failure; the remaining two patients had cerebral palsy. All of the patients were HIV-negative.

One hundred fifty-nine (97.5%) patients had received the BCG vaccine and 75 (46%) had a history of TB contact. The father was the most common source, at 50% of cases. Although drug-susceptibility results were not available for all patients, multidrug resistance was not encountered and the resistant cases had only isoniazid resistance.

### 3.2. Clinical and Laboratory Findings

Prolonged cough was the most common symptom (n: 80, 49%), followed by fever (n: 72, 44.1%), weight loss (n: 45, 27.6%), night sweating (n: 44, 26.9%), sputum production (n: 20, 12.2%), convulsion (n: 4, 2.4), and hemoptysis (n: 3, 1.8%).

Ninety-six (58.9%) patients had TST positivity, with a mean diameter of 13.47 ± 8.30 cm. Sixty-four patients (39.3%) had IGRA positivity. Of 163 patients, 34 had confirmed TB. Acid-resistant bacteria were observed in 34 (20.9%) body fluid samples, and culture positivity was observed in 33 (20.2%) samples. Six patients had isoniazid resistance.

The mean white blood cell count (WBC), hemoglobin, neutrophil count, lymphocyte count, platelet count, C-reactive protein (CRP) level, and erythrocyte sedimentation rate (ESR) were 12,885.82 ± 1867.94/mm^3^, 11.3 ± 1.5 g/dL, 6667.34 ± 4440.08/mm^3^, 2463 ± 1423/mm^3^, 340,030 ± 136,355/mm^3^, 4.04 ± 4.93 mg/dL, and 52.69 ± 33.35 mm/h, respectively.

The histopathological findings were confirmed in 48 (29.4%) patients, while 131 (80.4%) patients had radiologic confirmation. Mediastinal or hilar lymphadenopathy (41.1%) was the most common radiologic finding, followed by lobar consolidation (35.5%), patchy consolidation (21%), pleural effusion (11.1%), miliary TB (7.3%), and cavitary TB (3.0%). The radiologic findings of TB meningitis included tuberculum, basilar infiltration, and hydrocephalus.

Comparison of PTB, EPTB, and disseminated TB revealed that low socioeconomic status, TB contact, and low body weight were more common in disseminated TB (*p* = 0.000, 0.021, and 0.011, respectively) and TST positivity was more common in PTB (*p* = 0.012). Histopathologic confirmation was more common in EPTB (*p* = 0.000), while radiologic confirmation was more common in PTB (*p* = 0.000). Acid-resistant bacteria and culture positivity were more common in disseminated TB (*p* = 0.000 for both). The electrolyte sedimentation rate (ESR), CRP values, and platelet count were higher in disseminated TB (*p* = 0.006, 0.015, and 0.000, respectively). Comparison of patient characteristics between PTB, EPTB and disseminated TB are shown in Table 2.

### 3.3. Patient Treatment and Outcomes

Considering the clinical picture and risk factors, 112 (68.7%) patients received a treatment regimen containing four drugs, while 51 (31.3%) received a regimen containing three drugs. Streptomycin was the fourth drug in 86 (76.7%) patients and ethambutol was preferred in 26 (23.3%) patients. The mean length of treatment was 11.55 ± 5.33 months (9–24). During the course of treatment, 29 (18%) patients had a transient increase in alanine aminotransferase and aspartate aminotransferase levels and 11 (6.7%) had increased uric acid levels. We did not interrupt treatment. None of the patients developed hearing or visual impairment as a complication of the anti-TB treatment.

None of the patients died due to TB. Only one of the patients with TB meningitis had a minor neurologic disability. The patient with gastrointestinal TB developed an ileocecal perforation in the ninth month of treatment.

## 4. Discussion

Our study focused on the clinical, laboratory, and radiologic findings of PTB and EPTB in the pediatric age group. Pulmonary parenchymal disease and intra-thoracic adenopathy are the most common clinical manifestations of childhood TB worldwide [12,13]. The distribution of childhood TB in Turkey was also similar to those of previous reports, with 51% of cases PTB, 43% EPTB, and 4% disseminated TB [14]. In our study, 44.8% of the cases were PTB, 43.6% were EPTB, and 11.7% were disseminated TB, different from the previous reports. In the current study, the rate of EPTB was significantly higher than the 24.8% rate reported by Tsai et al. and similar to the 46% reported by Jain et al. [15,16]. As the lymphohematogenous spread is higher in children, the risk of dissemination and development of EPTB are higher in childhood compared to those in adults. However, the reported incidence of EPTB is generally low due to the variable clinical manifestations that can mimic other inflammatory or neoplastic diseases [3,17]. In the present study, the most frequent extrapulmonary TB form was TB lymphadenitis, followed by gastrointestinal TB. Lymphadenitis and meningitis are generally the most frequent TB forms; however, meningitis was the most common EPTB form in the study by Jain et al. [16].

The gender prevalence of TB varies in different reports; the percentage of female patients (57.7%) was higher in the current report, similar to Taiwanese study [15]. The mean age of the patients was 118.21 ± 60.44 months (range: 3–204), comparable to that (8.86 ± 4.19 years) in another report from our country [18]. The median age of TB patients was as young as 31 months in one study [15]. Young age is an important risk factor for the dissemination of TB; however, comparison of the possible risk factors for PTB and EPTB revealed no differences in age or gender.

TB is generally a disease of poor and malnourished populations. In the current report, 97 (59.5%) families had a low socioeconomic status. Low socioeconomic status was determined as families who work with minimum wage. The body weights of 80 patients (49.1%) were below the third percentile. Malnutrition and low socioeconomic status were important diagnostic variables in our study and both were more common in patients with disseminated TB (*p* = 0.011 and *p* = 0.000). Various reports evaluating the risk of malnutrition in parallel with the socioeconomic and nutritional status report it to play an important role in the dissemination and prognosis of TB infection [19,20].

The BCG vaccine, which is the first developed and currently being used vaccine, improves cellular immunity and protects against TB meningitis and the dissemination of TB. The vaccine prevents a large number of deaths caused by TB every year, as TB is the leading infectious cause of death worldwide [21,22,23]. BCG vaccination is strongly recommended for children in high-risk populations. The BCG vaccine is included in the national childhood immunization program in our country and 159 (97.5%) patients in the present study had received the BCG vaccine. Galli et al. reported that the vaccination status of children with latent TB was higher than that in children with active TB cases [24]. However, the vaccine does not prevent primary infection, latent, or reactivation TB. Jain et al. reported no significant difference in BCG vaccine status between children with and without TB, with an 88% vaccination rate [16]. Although TB developed in our patients despite the high vaccination rate, none of them died or developed severe complications.

Contact history is important for the diagnosis of TB, especially in countries with low endemicity. The incidence of contact history varies between 25–66% in various reports and is 16–23.7% in our country [25,26,27,28]. In the present study, 46% of patients had a contact history, with the father the most common infectious person and TB contact higher in disseminated TB. Wu and colleagues suggested that household contact has a higher risk of severe TB, with a reported rate of 31% [29].

In this study, 96 (58.9%) patients had TST positivity and 64 (39.3%) patients had IGRA positivity. TST positivity was more common in PTB (*p* = 0.012) and IGRA positivity did not differ significantly between groups. The TST is the major screening test in our country and the positivity rate was similar to that in other reports from our country [18,27]. However, the TST has limitations such as administration and interpretation technique and the vaccination, immune, and malnutrition status of the patient. The costs and technical demands are the main handicaps of the IGRA. While studies have reported the superiority of IGRA for the diagnosis of TB, WHO recommends that IGRA should not replace the TST in low- and middle-income countries for the diagnostic work-up of children. However, TB is a problem of populations with low socioeconomic status [30]. It is also necessary to remember that, while a positive result is helpful, a negative result for both tests does not rule out disease in young children.

Microbiological confirmation is important not only for definitive diagnosis but also for drug susceptibility tests to exclude drug-resistant TB. However, bacteriological confirmation is not always possible in children due to the technical difficulties in sampling and paucibacillary in children [2,5,6]. In the present study, we were able to perform microbiological evaluation in all of our patients. Acid-resistant bacteria were observed in 34 (20.9%) body fluid samples, and culture positivity was demonstrated in 33 (20.2%) samples. The mycobacterial culture positivity rate is 23.5–31% in our country whereas culture positivity rates are between 30% and 40% worldwide [5]. We were able to perform drug susceptibility tests in only 50 patients, six (12%) of which had isoniazid resistance. Studies on drug resistance in childhood TB are rare worldwide and in our country due to the fact that few centers are capable of performing drug susceptibility tests. However, the identification of resistance is important both for drug choice and duration of treatment; therefore, we hope to increase the feasibility of this test.

While the levels of ESR, CRP, and WBC vary in TB, there may also be signs of chronic inflammation such as anemia, hypoalbuminemia, and thrombocytosis. In the present study, the mean ESR, CRP value, and WBC count were elevated and the mean hemoglobin was decreased, with higher ESR, CRP value, and platelet count in disseminated TB (*p* = 0.006, 0.015, and 0.000, respectively).

The radiological features of pediatric PTB are extremely variable and there is usually no pathognomonic radiologic sign of TB. Mediastinal or hilar lymphadenopathy with parenchymal involvement, cavitation, pleural effusion, and calcifications are the most common signs. Mediastinal or hilar lymphadenopathy (41.1%) was the most common radiologic finding in this study, as was also reported by Boloursaz et al. [31]. Hilar lymphadenopathy was also the most common finding in a pediatric study from our country, which was followed by miliary pattern [27]. Miliary pattern was low in our patients, however, Boloursaz et al. did not demonstrate any miliary TB. The rate of cavitary TB was 3% in our patients, similar to literature which is reported as 1–7% [31,32].

Tuberculosis treatment can be difficult in children. The drugs are usually in tablet and capsule forms suitable for adults, only rifampicin has suspension formulation. So, the compliance and bioavailability of treatment can be low in children. However, children can tolerate high doses and the side effects are rare in children. Tuberculosis has good prognosis if properly treated. The treatment of TB must be a state policy and the government should encourage and provide financial support to patients for treatment.

Our study has several limitations. First, it was a single-center, retrospective study; thus, the results cannot be extrapolated. Second, the history of source case in terms of microbiological results and drug resistance was mostly insufficient. Finally, histological confirmation was available for only a few patients. Multicenter studies with larger sample sizes are necessary to verify the microbiological results and drug resistance findings.

## 5. Conclusions

This study was focused on the clinical, laboratory, and radiologic findings of children diagnosed with PTB and EPTB at a university hospital. We found out that low socioeconomic status, TB contact, and low body weight were more common in disseminated TB. Therefore, TB should be considered in patients admitted with different complaints and signs in populations with high TB incidence and low socioeconomic status. Contact history is also very important for accurate diagnosis, and anti-TB treatment should be initiated as soon as possible because of inadequate current data to establish a diagnosis.

We did not experience any major complications and none of the patients died due to TB, probably due to the high vaccination rate in our patients. There is a positive association between BCG vaccine efficacy and severe progressive TB and it is already evident that the BCG vaccine protects against miliary TB, complications, and disseminated TB.

## Figures and Tables

**Table 1 medicina-55-00428-t001:** Demographic findings of patients.

Patient Number	(n = 163) n (%)/Mean ± S.D.
Sex	
Male	69 (42.3%)
Female	94 (57.7%)
Diagnosis of patients	
Pulmonary TB	73 (44.8%)
Extrapulmonary TB	71 (43.6%)
Lymphadenitis	29 (32.2%)
Gastrointestinal TB	14 (15.6%)
TB Meningitis	12 (13.3%)
Miliary TB	12 (13.3%)
Pleural TB	10 (11.1%)
Bone TB	6 (6.7%)
Others	7 (7.7%)
BOTH	19 (11.7%)
The mean age of patients (years)	9.85 ± 5.03 years (0.25–17)
TB contact	75 (46%)
BCG vaccination	159 (97.5%)
Low socioeconomic status	97 (59.5%)
Low body weight	80 (49.1%)
Comorbid disorder	20 (12.3%)
TST positivity	96 (58.9%)
IGRA positivity	64 (39.3%)
ARB positivity	34 (20.9%)
Culture positivity	33 (20.2%)
Histopathologic confirmation	48 (29.4%)
Radiologic confirmation	131 (80.4%)

TB: tuberculosis; BCG: bacille Calmette–Guérin; TST: tuberculin skin test; IGRA: interferon-gamma release assay; ARB: acid-resistant bacteria.

**Table 2 medicina-55-00428-t002:** Comparison of patient characteristics between pulmonary tuberculosis (PTB), extrapulmonary tuberculosis (EPTB), and disseminated tuberculosis (TB).

	Pulmonary TB (n = 73)	Extrapulmonary TB (n = 71)	Disseminated TB (n = 19)	*p* Value
**Sex (male)**	35 (47.9%)	26 (36.6%)	8 (42.1%)	0.388
**Age (month)**	112.0 ± 56.0	126.9 ± 60.7	109.6 ± 73.8	0.273
**Low socioeconomic status**	49 (67.1%)	31 (43.7%)	17 (89.5%)	0.000
**BCG vaccination**	72 (98.6%)	68 (95.8%)	19 (100.0%)	0.413
**TB contact**	34 (46.6%)	27 (38.0%)	14 (73.7%)	0.021
**Low body weight**	43 (63.0%)	24 (33.8%)	13 (68.4%)	0.011
**Comorbid disease**	11 (15.1%)	5 (7.0%)	1 (5.3%)	0.213
**TST positivity**	52 (71.2%)	36(50.7%)	8 (42.1%)	0.012
**IGRA positivity**	33 (45.2%)	24(33.8%)	7 (36.8%)	0.365
**Histopathologic confirmation**	2 (2.7%)	38 (53.5%)	8 (42.1%)	0.000
**Radiologic confirmation**	73 (100%)	41 (57.7%)	17 (89.5%)	0.000
**Drug resistance**	1 (1.4%)	4 (5.6%)	2 (10.5%)	0.163
**ARB positivity**	5 (6.8%)	21 (29.6%)	8 (42.1%)	0.000
**Culture positivity**	5 (6.8%)	18 (25.4%)	10 (52.6%)	0.000
**Sedimentation**	53.8 ± 34.7	46.0 ± 30.1	73.3 ± 32.0	0.006
**C-reactive protein**	3.9 ± 4.6	3.4 ± 3.6	7.0 ± 8.6	0.015
**White blood cell**	12,341.9 ± 20,320.3	13,510.0 ± 19,277.9	12,643.2 ± 6370.6	0.931
**Neutrophil number**	5924.5 ± 4135.8	6920.7 ± 4461.0	8573.7 ± 5028.9	0.055
**Lymphocyte number**	2328.8 ± 1157.6	2526.3 ± 1479.9	2750.5 ± 2044.3	0.460
**Platelet number**	308,575.3 ± 106,264.2	337,450.7 ± 114,472.2	470,526.3 ± 221,798.3	0.000
**Hemoglobin value**	11.6 ± 1.3	11.3 ± 1.7	10.8 ± 1.4	0.109

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
