# Peer review of "Evaluation of Clinical and Laboratory Characteristics of Children with Pulmonary and Extrapulmonary Tuberculosis"

_medicina, 2019, doi:10.3390/medicina55080428_

Round 1
Reviewer 1 Report
p.p1 {margin: 0.0px 0.0px 0.0px 0.0px; font: 12.0px Times} p.p2 {margin: 0.0px 0.0px 0.0px 0.0px; font: 12.0px Times; min-height: 14.0px}The manuscript by Aygun et al. presents the retrospective analysis of the cohort of children diagnosed with TB over the 10 years period at one hospital in Istanbul, Turkey. Due to technical and ethical issues, childhood TB is not studied in as much depths as adult TB and requires extensive further studies. Therefore, the manuscript might be of interest for TB specialists and paediatricians especially those working in high TB incidence settings.
However, I have a number of comments regarding the text of the manuscript which are as follows:
Abstract
Line 17: you missed word “TB” I believe
Introduction
Lines 25-27: please revise the sentence. What ‘both’ refers to?
Lines 34-40: in fact, TB incidence is falling at about 2% every year (please see WHO TB Report 2018, p.27). The mortality rate is also reducing (1.3 million TB deaths in 2016). TB incidence in Turkey in 2017 is estimated by WHO as 17 per 100,000 inhabitants. Please provide the latest data from the above-mentioned report.
Lines 48-52: the aim of the study is not shown to be logically linked to the previous paragraph. Could you please provide more clear reasons for conducting the study?
Material and method
Lines 75-77: please check these sentences, why are they repeating each other and how did you classify PTB+EPTB? Here you use the term ‘miliary TB’ while in other parts of the manuscript you call it ‘disseminated TB’. These two forms of TB give different histological pictures as it was pointed out by your colleagues from Istanbul University (Mert A, Ozaras R. A terminological controversy: do disseminated and military tuberculosis mean the same? Respiration. 2005 Jan-Feb;72(1):113). Please use one more appropriate term throughout the text.
Lines 86-89: was the full set of laboratory tests you describe performed on each sample?
Results
Line 107: please check the sentence: “There included…”
Line 111: 10 patients with PTB were included in EPTB cases? It is not clear.
Lines 111-113: what patients do you mean here? What does “Seven of (7.7%) three…” mean? Please revise the sentence.
Line 115: you mention 17 patients here and 20 patients – in Table 1.
Line 127: please add % to the number
Line 129: you say 96 patients with TST positivity here and 98 – in Table 1.
Table 2: the title might be misleading since you compare 3 groups, not 2. Moreover, you compare patient characteristics, not 2 types of TB.
Discussion
Line 185: perhaps ‘no differences IN age or gender’?
Line 209: perhaps ‘had TST positivity’?
Lines 225-227: you can’t compare culture positivity rate with AFB smear positivity.
Author Response
1. Tuberculosis (TB) was added in two parts in line 17.
2. ‘’Both’’ was revised as all three parameters.
3. Sorry for our mistake abd the old litherature, we revised according to latest data.
4. We changed the introduction and tried to provide the reason of our study. There was a confusing section, we tried to write it again.
5. You are right, there is a confusing terminology there, disseminated TB was called as both pulmonary and extrapulmonary TB, but miliary TB is a histological diagnose and has miliary pattern.
6. We have special microbiology department only for TB in our clinic and a specialist studies the samples. Each samples were examined.
7. We corrected the confusing sentence.
8. We corrected the mistake, it was pleural TB, sorry for it.
9. We correted the confusing sentence, seven was referred as others composed of renal, skin, genitourinary and pericardial TB.
10. We corrected the numbers, sorry for it.
11. Numbers of patients were added to % in clinical and laboratory findings.
12. We corrected the numbers, sorry for it.
13. We corrected the misleading sentence of Table 2.
14. The word ‘’in’’ was added.
15. The Word ‘’positivity’’ was added.
16. We corrected the sentence according to culture positiviy and removed ARB positivity.
Reviewer 2 Report
The information provided here is important but does not add much to already published literature. This is due to small sample size and lack of clinical data. The author did not provide information on drug resistance status of each infection. It is also interesting that other than isoniazid no other drug resistance was observed in any patient. The authors only included patients that were hospitalized. Not all patients with pulmonary TB are hospitalized. I am not sure what authors mean here. Also, Please cite appropriate reference here. “Childhood TB is a health index…. Future TB pool.”
1. Line 46 “The microbiological confirmation rate is also low in children” I am not sure if this is correct. Please explain and cite related reference.
2. Line 108 “mean age was…. (3-204)” Please express data in years. Is variation shows as SD or SEM.
3. How many patients had lung granuloma/cavitation. ?
4. What was the treatment length
5. Line 114 What is definition of low socioeconomic status ?
6. The authors should discuss the effect of BCG vaccination, EPTB risk factors and clinical correlates of TB in more details. (PMID: 28137237,30135583 )
7. The authors need to plot the data with time so readers can understand how data was collected in 10 year period and also plot the clinical correlates with time.
8. There are issues with reference formatting.
Author Response
Childhood TB is a health index that reflects uncontrolled adult TB in the population. Childhood TB is also an important epidemiological indicator in terms of forming the future TB pool. This sentence was written by our professor Yıldız Camcıoğlu, she is a specialist in childhood TB, so we could not find a reference, if you think it is not appropriate, we can remove it.
1. The microbiological confirmation rate is low in children because of technical difficulties. We corrected the sentence according to that, sorry for confusing sentence.
2. We corrected the mean age according to years.
3. We added the rate of cavitary TB, sorry for our mistake.
4. We added the treatment length, we forgot it.
5. Low socioeconomic status was determined as families who work with minimum wage in our country, generally only the fathers work and mothers are house wife.
6. We tried toexplain the effect of BCG vaccination and extrapulmonary TB.
7. We have tuberculosis out patient clinic and we have well-maintained data of our patients. The detailed data of 163 patients diagnosed with TB was analyzed. We menitioned about it in material and method section.
8. We revised the reference formatting.
Reviewer 3 Report
In this manuscript, authors evaluated medical records of children in Turkey with a diagnosis of tuberculosis (both PTB and EPTB), a worldwide public health problem. Results show that socioeconomic status and malnutrition are important diagnostic variables.The manuscript is well written and is suitable for publication in Medecina in present form.
Minor suggestions:
-Authors affiliations are missed in the manuscript
-In the discussion, it could be interesting to compare the results of the study with the report of other papers (in other country, in Europe,… if it existe). Some of them are described in the manuscript but it could be nice to add in the manuscript where the study was conducted.
-In the conclusion, you could add some words to say that it is very important to adapt the patient's treatment based on these observations and that faster and cheaper diagnostic techniques are needed.
Author Response
Thank you for your nice criticism. We revised the paper according to your suggestions. Changes are shown as colored text in the revised manuscript.
We tried to add litherature that compares the results of childhood TB from our country and Europe.
We tried to write a paragraph about the treatment and adaptation of patients.
Round 2
Reviewer 1 Report
p.p1 {margin: 0.0px 0.0px 0.0px 0.0px; font: 12.0px Times}Most of my comments have been addressed. Overall, the manuscript has been improved following the reviewers' comments.
Still, there are a few minor comments that I would advise to address before making a decision on accepting the manuscript for publication.
Line 113: you revised the sentence but still please correct the following: ‘Seven of (7.7%) of them…’
Line 118: you changed the number to 20 patients but in the following 2 sentences you describe only 17 patients. Please correct.
Line 131: please add % to the number 2.4.
Line 244: probably it’s miliary pattern, not military.
Author Response
Dear Reviewer
Thank you for your nice criticism. We revised the paper according to your suggestions. Changes are shown as colored text in the revised manuscript.
